# Cost-Free Personalization via Information-Geometric Projection in Bayesian Federated Learning

**Nour Jamoussi**                                                                *nour.jamoussi@eurecom.fr*
*Communication Systems Department*
*EURECOM, France*

**Giuseppe Serra**                                                               *giuseppe.serra@eurecom.fr*
*Communication Systems Department*
*EURECOM, France*

**Photios A. Stavrou**                                                           *fotios.stavrou@eurecom.fr*
*Communication Systems Department*
*EURECOM, France*

**Marios Kountouris**                                                              *mariosk@ugr.es*
*Department of Computer Science and Artificial Intelligence, University of Granada, Spain*
*Communication Systems Department, EURECOM, France*

**Reviewed on OpenReview:** *https://openreview.net/forum?id=9yOjCrxjDR*

## Abstract

Bayesian Federated Learning (BFL) combines uncertainty modeling with decentralized training, enabling the development of personalized and reliable models in the presence of data heterogeneity and privacy constraints. Existing approaches typically rely on Markov Chain Monte Carlo (MCMC) sampling or variational inference, often incorporating personalization mechanisms to better adapt to the local data distributions. In this work, we propose an information-geometric projection framework for personalization in parametric BFL. By projecting the global model onto a neighborhood of the user's local model, our method enables a tunable trade-off between global generalization and local specialization. Under mild assumptions, we show that this projection step is equivalent to computing a barycenter in the statistical manifold, allowing us to derive closed-form solutions and achieve cost-free personalization. We apply the proposed approach within a variational learning setup using the Improved Variational Online Newton (IVON) optimizer and extend it to general aggregation schemes in BFL. Empirical evaluations under heterogeneous data distributions confirm that our method effectively balances global and local performance with minimal computational overhead. Code is available at `https://github.com/NourJamoussi/Information-Geometry-for-Bayesian-Federated-Learning`.

## 1 Introduction

Federated learning (FL) is a collaborative machine learning paradigm designed to preserve data privacy. By connecting a central server to multiple participants, commonly referred to as clients or end-users, FL enables distributed model training while keeping data local to each client. In a typical FL setting, clients train local models on their private datasets, and the server aggregates these locally updated models into a global model. After each round of local updates, clients share their updated model parameters with the server, which aggregates them to refine the global model. The updated global model is then sent back to the clients for further local training McMahan et al. (2017).

In real-world federated learning scenarios, statistical heterogeneity among clients is a pervasive challenge. For instance, variations in Magnetic resonance imaging (MRI) scanners across hospitals can induce *feature skewness*, where the input data distributions differ substantially between clients. Likewise, *label distribution shifts* may occur when certain clients have disproportionately more samples from specific classes than others, such as in specialized hospitals focusing on particular diseases compared to general hospitals. This heterogeneity violates the standard assumption of independent and identically distributed (i.i.d.) data, leading to degraded performance and increased uncertainty of the global model when evaluated on local test data.

Personalized Federated Learning (PFL) is a paradigm in which clients adapt the global model to their local data, i.e., perform model personalization, with the goal of mitigating the impact of statistical heterogeneity. In Tan et al. (2023), the authors propose a comprehensive taxonomy of PFL approaches, classifying them into two main categories: global model personalization techniques and personalized model learning techniques. The former category includes methods such as client selection Yang et al. (2021), meta-learning Fallah et al. (2020), and transfer learning Chen et al. (2020), while the latter encompasses strategies such as parameter decoupling Arivazhagan et al. (2019), knowledge distillation Jeong & Kountouris (2023), and clustering-based approaches Sattler et al. (2020). Furthermore, several studies Zhang et al. (2022b); Boroujeni et al. (2024); Kotelevskii et al. (2022); Zhu et al. (2023) have applied Bayesian methods within the PFL framework, demonstrating improved model calibration and uncertainty quantification. The transition from deterministic FL to Bayesian FL (BFL) naturally motivates an investigation of the manifolds to which clients' and global posterior distributions belong. This perspective enables the exploration of their geometric properties and the definition of meaningful operations on these manifolds, as discussed in Jamoussi et al. (2024).

In this work, we study the PFL paradigm through the lens of Bayesian learning, leveraging principles from Information Geometry to develop a novel personalization method for BFL. Unlike existing approaches, the proposed method does not require fine-tuning, additional training, or access to either local or shared global data, making it a fully private and computationally cost-free personalization technique. By specifying a divergence metric, we obtain personalized model posteriors by projecting the global posterior distribution onto a sphere centered at the local posterior distribution, where the radius encodes the desired degree of personalization specified by the end-user. This formulation enables unrestricted personalization flexibility, allowing users to seamlessly control the trade-off between local adaptation and global generalization without incurring additional computational or privacy costs.

We show that, for any divergence function that is convex in its first argument, the projection problem is equivalent to computing the weighted barycenter between the local and global posterior distributions. This equivalence establishes a conceptual bridge between two fundamental concepts in Information Geometry, geometric projection and geometric barycenter, thereby providing an interpretable relationship between the radius of the projection sphere and the weights in the barycentric aggregation. Consequently, the degree of personalization becomes both intuitive and directly controllable by end-users.

The paper is organized as follows. Section 2 reviews the related work. Section 3 introduces the parametric BFL framework based on variational learning for local training and posterior aggregation to estimate the global posterior model. We extend the approach of Pal et al. (2024) by employing the Improved Variational Online Newton (IVON) optimizer Shen et al. (2024) for multiple aggregation methods. In Section 4, we present a novel personalization strategy for global BFL models using information-geometric projection and show its equivalence to barycentric aggregation for divergence functions convex in their first argument. The proposed method is general and applicable beyond parametric BFL, including non-parametric settings. For the parametric case, we exploit closed-form barycentric solutions for specific divergences, yielding significant computational gains. Section 5 reports the experimental results, comparing aggregation techniques under the variational learning framework and benchmarking our method against state-of-the-art approaches. Finally, Section 6 discusses the broader applicability of the proposed method to continual learning.

## 2 Background and Related Work

**Bayesian Federated Learning.** BFL seeks to integrate the advantages of Bayesian Deep Learning, such as improved model calibration and uncertainty quantification, within the FL framework. Following the classification proposed in Cao et al. (2023), BFL approaches can be broadly categorized into two main types:

client-side BFL Zhang et al. (2022b); Zhu et al. (2023); Liu et al. (2023); Boroujeni et al. (2024); Hasan et al. (2024); Bhatt et al. (2024) and server-side BFL Corinzia et al. (2019); Chen & Chao (2020); Al-Shedivat et al. (2020); Guo et al. (2023), with certain overlaps between these groups, reflecting the diversity in model formulations and inference techniques. In Jamoussi et al. (2024), the authors extend the discussion of the parametric client-side BFL framework by categorizing methods according to the techniques used to aggregate local posteriors and propose a unified framework for aggregation rooted in information geometry.

However, within the scope of this work, it is useful to introduce an additional categorization of BFL based on the adopted modeling assumptions, distinguishing between parametric and nonparametric Bayesian methods.

- *Parametric BFL*: Parametric Bayesian learning provides a principled framework within Bayesian statistics, wherein the data generation process is assumed to follow a parametric model. Specifically, this approach assumes that the data distribution can be fully characterized by a finite set of parameters. The parametric family most commonly considered in BFL is the Gaussian family, in which both client-specific posteriors and the global model posterior are approximated using Gaussian distributions Zhang et al. (2022b); Ozer et al. (2022); Guo et al. (2023); Kim & Hospedales (2023); Pal et al. (2024); Swaroop et al. (2025). This Gaussian assumption facilitates tractable inference and efficient parameter aggregation.

- *Nonparametric BFL*: Unlike parametric BFL, nonparametric Bayesian learning does not restrict the model to a specific family of distributions. Instead, it provides the flexibility to learn directly from the data, allowing the structure and complexity to emerge naturally rather than being predefined. This approach leverages distributions over infinite-dimensional function spaces, such as Gaussian Processes or Dirichlet Processes, as employed in the context of BFL in Yurochkin et al. (2019). Additionally, particle-based variational inference, used in Kassab & Simeone (2022) for BFL, aligns with the nonparametric perspective by using a set of particles to approximate posteriors without assuming a predefined parametric family.

**Personalized Bayesian Federated Learning.** Personalized BFL extends the standard BFL framework by incorporating client-specific adaptations into the shared global model. FedPop Kotelevskii et al. (2022) achieves personalization by modeling each client's data generation process as a combination of fixed shared population parameters, which describe the common data model, and client-specific random effects, which capture heterogeneity in client data, providing personalization and uncertainty estimation capabilities. To efficiently infer these parameters, FedPop employs MCMC methods to approximate local posterior distributions and perform stochastic optimization in a federated setting. Although MCMC enables flexible and asymptotically exact Bayesian inference, it can be computationally expensive in large-scale scenarios. To address this, pFedBayes Zhang et al. (2022b) adopts variational inference to approximate client-specific posteriors, optimizing a tractable Evidence Lower Bound (ELBO) to achieve computational efficiency in BFL. Although pFedBayes presents a promising approach to personalization in BFL, certain aspects of its methodology introduce notable limitations. First, the degree of personalization is uniformly applied across all clients and determined empirically, without an adaptive mechanism or an intuitive rationale for tailoring it to individual client characteristics. Second, it defines the local model as personalized based on the use of variational inference, where the global model serves as a prior within the KL divergence term of the ELBO. This formulation may be viewed as controversial within the Bayesian community, since the classical interpretation of a prior belief assumes independence from the observed data. More recently, pFedVEM Zhu et al. (2023) has leveraged variational inference to estimate the client-specific uncertainty and model deviation by modeling client parameters as Gaussian distributions centered around a global latent variable. It employs a confidence-based aggregation strategy that ensures that clients with lower uncertainty and smaller deviations contribute more to the global model.

Unlike the aforementioned methods, our approach introduces personalization as an additional step independent of the training process, requiring only access to the global and local posteriors, with no need for extra data or fine-tuning. Moreover, our method provides access to three different variants of the model, i.e., local, personalized, and global, as suggested in Divi et al. (2021), with the ability to continually adjust the personalized variant without incurring additional training costs.

**Information Geometry and Optimal Transport in FL.** Information geometry Amari (2016) provides a general framework for studying the geometric properties of statistical manifolds through divergence functions, allowing distributions to be intuitively treated as elements of a manifold. This framework naturally extends fundamental geometric concepts, such as projections onto constraint sets Csiszár (1975); Csiszár & Matus (2003) and the identification of barycenters, which serve as statistical centers of mass for a given set of probability distributions Nielsen & Nock (2009); D'Ortenzio et al. (2022).

In contrast, optimal transport Villani (2009) focuses on identifying the most efficient transport plan between two probability distributions with respect to a given cost function. Under specific assumptions on the cost function, optimal transport enables the definition of distance functions on the statistical manifold, as exemplified by the family of Wasserstein-p distances Villani (2009). The Sinkhorn algorithm Cuturi (2013) serves as a natural bridge between information geometry and optimal transport, enabling efficient computation of transport plans by incorporating entropy regularization into the optimal transport problem.

Multi-Marginal Optimal Transport (MMOT) Gangbo & Święch (1998); Pass (2015) extends the classical two-marginal optimal transport problem to multiple distributions by seeking an optimal joint coupling that minimizes a given $n$-ary cost function over all marginal distributions. Furthermore, for specific forms of the $n$-ary cost, the MMOT problem can be shown to be equivalent to computing the Wasserstein barycenter (WB) of the set of marginals. This equivalence highlights MMOT as a more general framework within which the WB emerges as a special case, characterizing the optimal interpolation of multiple distributions in Wasserstein space. Recently, applications of concepts from information geometry and optimal transport have been explored in the FL setting. In Farnia et al. (2022), the authors introduce FedOT, a PFL algorithm that integrates optimal transport with model training. FedOT employs deterministic FL, leveraging MMOT to align data distributions. Specifically, it learns transport maps that transform data points from heterogeneous distributions into a shared domain while simultaneously training a predictive model on the mapped data. Similarly, FedDRO Li et al. (2024) aggregates client data distributions via a Wasserstein barycenter and trains the global model against worst-case perturbations within a Wasserstein ball centered at this barycenter. In the context of BFL, Hassan et al. (2023) considers structured latent variable models in which local latent variables are kept private, formulating a decentralized variational inference problem and proposing a communication-efficient aggregation scheme based on Wasserstein barycenters. Similarly, Jamoussi et al. (2024) introduces a unifying framework for aggregation in BFL grounded in barycentric aggregation.

## 3 Parametric Bayesian Federated Learning

### 3.1 Learning Phase

The primary objective of BFL is to estimate the posterior distribution of the global model parameters, denoted as $p(\theta^*|\mathcal{D})$, using the posterior distributions of local models, $p(\theta_k|\mathcal{D}_k)$. However, exact posterior inference is typically computationally intractable, necessitating the use of approximate inference methods. In this study, we adopt variational learning to approximate local posterior distributions based on a shared prior distribution $p(\theta)$ and client-specific likelihoods $p(\mathcal{D}_k|\theta_k)$. Given a parametric distribution family $\mathcal{Q}$, optimization seeks a distribution $q \in \mathcal{Q}$ that minimizes the KL divergence from the true posterior distribution $p(\theta|\mathcal{D})$, i.e.,

$$\min_{q(\theta) \in \mathcal{Q}} D_{\mathrm{KL}}(q(\theta)\|p(\theta|\mathcal{D})). \tag{1}$$

However, direct optimization of equation 1 is generally intractable, motivating the use of the Negative Evidence Lower Bound (ELBO) as a surrogate objective:

$$\min_{q(\theta) \in \mathcal{Q}} -\mathbb{E}_{q(\theta)}[\log p(\mathcal{D}|\theta)] + D_{\mathrm{KL}}(q(\theta)\|p(\theta)). \tag{2}$$

We approximate the posterior distributions of the local models $p(\theta_k|\mathcal{D}_k)$, $\forall k \in \{1, \ldots, N\}$ by optimizing the objective in equation 2 using the IVON optimizer Shen et al. (2024), which is grounded in the Bayesian learning rule Khan & Rue (2021). Unlike classical variational inference, IVON integrates variational learning directly into the optimization process, without requiring modifications to the model architecture or loss

function. IVON maintains a Gaussian posterior over the weights through efficient second-order updates based on reparameterized Hessian estimates, enabling uncertainty-aware learning at a computational cost comparable to deterministic training with Adam. This makes Bayesian deep learning more scalable in practice. Subsequently, the local posteriors are aggregated to obtain the global posterior distribution $p(\theta^*|\mathcal{D})$.

Based on this formulation, we introduce the following assumptions regarding the common prior $p(\theta)$ and the variational family $\mathcal{Q}$, which will hold throughout the experimental setting described in Section 5.

**Assumption 1** (Mean-field Model)**.** *The variational family $\mathcal{Q}$ consists of d-dimensional Gaussian distributions with independent marginals. Specifically, $\theta \sim \mathcal{N}(\mu, \Sigma)$, where $\mu \in \mathbb{R}^d$ is the mean vector and $\Sigma = \mathrm{diag}(\sigma_1^2, \ldots, \sigma_d^2)$ is a diagonal covariance matrix.*

### 3.2 Aggregation Phase

The aggregation phase refers to the process of combining locally trained model updates from multiple clients into a single global model. In what follows, we outline and discuss several aggregation techniques for the $N$ local distributions, each consistent with Assumption 1 and parametrized with $\{(\mu_k, \Sigma_k)\}_{k=1}^N$.

- *Empirical Arithmetic Aggregation*, also known as naive aggregation, is employed in Zhang et al. (2022b); Ozer et al. (2022); Bhatt et al. (2024); Fischer et al. (2024). It computes the weighted average of the distribution statistics:

$$\Sigma_{\mathrm{EAA}} = \sum_{k=1}^N w_k \Sigma_k, \quad \mu_{\mathrm{EAA}} = \sum_{k=1}^N w_k \mu_k. \tag{3}$$

- *Barycentric Aggregation*, adopted for BFL in Jamoussi et al. (2024), uses the barycenter of the clients' posteriors as the aggregated model, minimizing the average discrepancy among client distributions. Some examples include:

  - *Wasserstein-2 Barycenter*: minimizes the average discrepancy in terms of the Wasserstein-2 distance between the clients' distributions. Under Assumption 1, a closed-form solution for the Wasserstein barycenter is derived in Álvarez-Esteban et al. (2016):

$$\Sigma_{\mathrm{W}_2^2} = \left( \sum_{k=1}^N w_k \Sigma_k^{\frac{1}{2}} \right)^2, \quad \mu_{\mathrm{W}_2^2} = \sum_{k=1}^N w_k \mu_k. \tag{4}$$

  - *Reverse KL Barycenter*: minimizes the average reverse KL divergences between the local distributions and coincides with the multiplicative aggregation of posteriors proposed in Al-Shedivat et al. (2020); Liu et al. (2023); Guo et al. (2023); Pal et al. (2024). Its closed-form expressions are given in Koliander et al. (2022):

$$\Sigma_{\mathrm{RKL}} = \left( \sum_{k=1}^N w_k \Sigma_k^{-1} \right)^{-1}, \quad \mu_{\mathrm{RKL}} = \Sigma_{\mathrm{RKL}} \sum_{k=1}^N w_k \Sigma_k^{-1} \mu_k. \tag{5}$$

To support the different aggregation strategies that operate on covariance matrices, and to avoid arbitrarily setting the effective sample size parameter in IVON, which ideally corresponds to the total dataset size but is often unknown at the server, we adopt a subclass of IVON that explicitly stores the covariance matrix and performs sampling directly from it, rather than relying on the Hessian matrix. This reformulation is made possible by the relation derived in Shen et al. (2024)

$$\sigma^2 = \frac{1}{N(h + \delta)}, \tag{6}$$

which links the variance $\sigma^2$ to the Hessian approximation $h$[1], the dataset size $N$, and the weight decay term $\delta$. In practice, the covariance matrix is first computed locally after estimating the Hessian. These

---

[1]For a multivariate Gaussian, the Hessian of the negative log-likelihood is proportional to the inverse covariance matrix. When the covariance is diagonal, the Hessian is also diagonal, so the Hessian approximation reduces to per-parameter scalar entries.

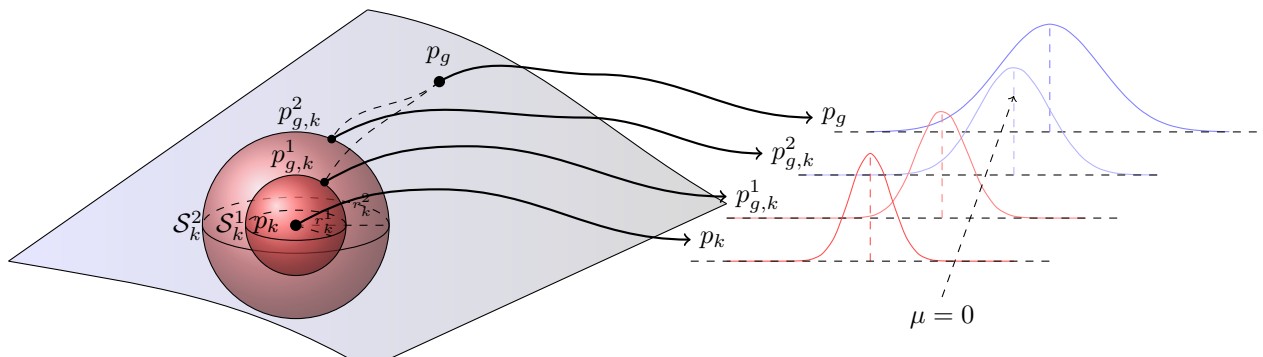

Figure 1: Personalization through information-geometric projection. The figure presents two projection scenarios illustrated with two local spheres $\mathcal{S}_k^1$ and $\mathcal{S}_k^2$ of increasing radius $r_k^1$ and $r_k^2$, highlighting the impact of the radius on the closeness of the projected distribution to the global or local distribution.

local covariance matrices are then aggregated at the server, and the resulting global covariance is distributed back to the clients, where it is used to reconstruct the updated local Hessians. Unlike prior work Pal et al. (2024), which employed IVON but was limited to RKLB-based aggregation due to its direct dependence on the Hessian, our formulation enables a broader range of covariance-based aggregation strategies.

## 4 Personalization via Information-Geometric Projection

The main contribution of this work lies in interpreting the personalization problem as an *information-geometric projection problem*. Given a divergence function $D$, the objective is to project the global posterior $p_g$ onto the projection set of the $k^{th}$ client, defined as a sphere centered at the local posterior $p_k$ with radius $r_k$. The projection is performed after FL training has been completed. Conceptually, this projection can be viewed as identifying the distribution within the "local neighborhood" of the client's posterior that best aligns with the global model. The radius $r_k$ quantifies the degree of personalization required by client $k$, where smaller values of $r_k$ correspond to stronger adherence to the local posterior, while larger values allow greater influence from the global model, as shown in Figure 1. In the following, we formally define the local sphere $\mathcal{S}_k$ induced by the divergence function $D$.

**Definition 1.** *(Local Sphere $\mathcal{S}_D(p, r)$) Given a statistical manifold $\mathcal{M}$ and a divergence function $D$, the local sphere $\mathcal{S}_D(p, r)$ centered at $p \in \mathcal{M}$ with radius $r \in [0, \infty)$ is defined as the set*

$$\mathcal{S}_D(p, r) = \{q \in \mathcal{M} : \quad D(q\|p) \leq r\} \subseteq \mathcal{M}.$$

To simplify the notation, we denote by $\mathcal{S}_k = \mathcal{S}_D(p_k, r_k)$ the local sphere associated with the $k^{th}$ client, centered at the local posterior $p_k$ with radius $r_k$. We are now ready to formally define the personalization problem as a projection problem.

**Problem 1.** *(Projection onto $\mathcal{S}_k$) Given a statistical manifold $\mathcal{M}$, a divergence function $D$, a global posterior $p_g \in \mathcal{M}$, and a local sphere $\mathcal{S}_k$, we define the projection of $p_g$ onto $\mathcal{S}_k$ as the optimization problem*

$$\min_{p \in \mathcal{S}_k} D(p\|p_g). \tag{7}$$

*The solution to this projection problem is identified as the personalized posterior distribution of client $k$, denoted by $p_{g,k} = \arg\min_{p \in \mathcal{S}_k} D(p\|p_g)$.*

**Remark 1.** *By varying the radius $r_k \in [0, D(p_g\|p_k)]$, the solutions of the associated projection problem trace the geodesic between $p_k$ and $p_g$, i.e., the shortest possible path connecting the two distributions under the geometry induced by the divergence $D$.*

To support our derivation, we first introduce the definition of a barycenter with respect to a divergence $D$.

**Definition 2.** *(D-barycenter) Given a statistical manifold $\mathcal{M}$, a divergence function $D$, and a set of distributions $\{p_k\}_{k=1}^N \subseteq \mathcal{M}$ with associated normalized weights $\{w_k\}_{k=1}^N$, the barycenter of the set $\{p_k\}_{k=1}^N$ is defined as*

$$p_D^*(\{p_k\}_{k=1}^N, \{w_k\}_{k=1}^N) = \arg\min_{q \in \mathcal{M}} \sum_{k=1}^N w_k D(q||p_k). \tag{8}$$

The following mild assumption is essential for establishing the equivalence between the projection and barycenter formulations, as shown in Theorem 1.

**Assumption 2.** *The divergence metric is a convex function in its first argument.*

**Remark 2.** *Most divergences used in practice, such as the family of $f$-divergences and the Wasserstein-p distances, are convex in both arguments, and therefore satisfy Assumption 2.*

**Theorem 1.** *Under Assumption 2, the solution of the projection problem 1 is equivalent to that of the weighted barycenter problem 2, i.e.,*

$$p_{g,k} = p_D^*(\{p_g, p_k\}, \{w_g, w_k\}), \tag{9}$$

*where the weights $w_g$ and $w_k$ are given by*

$$w_g = \frac{1}{\lambda + 1}, \quad w_k = \frac{\lambda}{\lambda + 1} \tag{10}$$

*for some $\lambda \in [0, \infty)$.*

The proof of Theorem 1 is provided in Appendix A.

We highlight the following observations regarding the relationship between $r_k$ and $\lambda$.

**Remark 3.** *There exists an inverse proportional relationship between the Lagrangian multiplier $\lambda$ and the radius $r_k$. Specifically, as $\lambda \to 0$, the radius $r_k \to \infty$, corresponding to the case where the personalized posterior coincides with the global posterior. Conversely, in the limit $\lambda \to \infty$, the radius $r_k$ vanishes ($r_k \to 0$), implying that the personalized posterior collapses to the local posterior.* **Hence, by selecting a value of $\lambda$, we implicitly determine the personalization radius $r_k$.**

The main advantage of the equivalence between projections and barycenters lies in the improved tractability of the barycentic formulation. For instance, under Assumption 1, analytical solutions are available for both the RKL divergence and the Wasserstein-2 distance, as shown in Koliander et al. (2022); Álvarez-Esteban et al. (2016). These closed-form expressions enable a straightforward and computationally efficient personalization procedure, incurring virtually no additional cost.

## 5 Experiments

### 5.1 Experimental Setting

**Datasets and Heterogeneity Simulation.** Interpretability and reliable uncertainty quantification are crucial in high-stakes domains such as banking and healthcare. When these institutions participate in FL, they typically do so in a cross-silo setting, where the number of clients is small but each holds a large local dataset. Reflecting this application focus, we concentrate our empirical evaluation on cross-silo simulations rather than cross-device benchmarks Caldas et al. (2018). To simulate these conditions, we evaluate our approach on three widely used image classification benchmarks: FashionMNIST, SVHN, and CIFAR-10, and we induce label shift across 10 clients using Dirichlet-based partitioning. Following prior works Li et al. (2020); Yurochkin et al. (2019); Wang et al. (2020a;b); Lin et al. (2020); Ozer et al. (2022), we sample client-specific label distributions from a Dirichlet distribution with concentration parameter $\beta = 0.5$, which induces label shift and yields non-i.i.d. data across clients.

**Metrics.** Following the taxonomy of evaluation metrics in Table IV of Tan et al. (2023), our evaluation focuses mainly on model performance and trustworthy AI criteria. For model performance, we report overall global-model accuracy as well as the average and variance of personalized-model accuracies, together with the worst-performing 10% of clients' accuracy. The latter two also connect to the fairness column in Table IV, and we elaborate on this in Section 5.5. In addition, our Bayesian formulation explicitly targets uncertainty quantification and model calibration, which are not explicitly listed among the model-performance or trustworthy-AI metrics in Tan et al. (2023). We therefore report Expected Calibration Error (ECE) and Negative Log-Likelihood (NLL) alongside accuracy-based measures to assess both the quality and the reliability of the predictive distributions. In contrast, we do not report system-performance metrics such as fault tolerance or system scalability, since our method has similar per-round complexity to FedAvg (see Section 3), and our focus here is on the statistical behavior of the personalized posteriors rather than on system-level aspects.

**Models Architecture and Hyperparameters.** Our model architecture consists of two convolutional layers with 5×5 kernels and ReLU activations, each followed by a 2×2 max-pooling layer. The extracted features are flattened and passed through three fully connected layers of sizes 120, 84, and 10, respectively, to produce the final class logits. For all state-of-the-art methods we compare against, we closely follow the implementation details provided in the original papers. The complexity of our architecture is comparable to that of the models used in these methods, ensuring a fair and consistent comparison. Whenever possible, we use the official codebases for implementation; otherwise, we carefully reproduce the setups following the authors' guidelines to maintain alignment with their reported settings. We make an exception for pFedBayes on FashionMNIST, where, instead of the original multi-layer perceptron with one hidden layer, we use our convolutional architecture to enable a fair comparison.

It is worth noting that training with IVON can be sensitive to hyperparameter settings, particularly to the initialization of the Hessian. Therefore, we report in Table 1 the hyperparameters used for training on each dataset. Here, $N_k$ denotes the size of the training dataset for the $k^{th}$ client.

Table 1: IVON Hyperparameters.

| params | FashionMNIST | SVHN | CIFAR-10 |
|---|---|---|---|
| initial learning rate | 0.1 | 0.1 | 0.1 |
| final learning rate | 0.01 | 0.01 | 0.01 |
| weight decay | 2e-4 | 2e-4 | 2e-4 |
| batch size | 64 | 64 | 64 |
| ESS | $N_k$ | $N_k$ | $N_k$ |
| initial hessian ($h_0$) | 5 | 2 | 1 |
| MC sample while training | 1 | 1 | 1 |
| MC sample while testing | 10 | 10 | 10 |

**Personalization Step.** For personalization, we restrict our experimental evaluation to RKLB and WB, as both barycenters preserve the Gaussian structure of the distributions. This preservation is a critical design choice, as it facilitates the adaptability of the personalized models. In particular, it ensures that the posterior distribution of the personalized model remains consistent with those of both the global and local models, which are assumed to be Gaussian under the variational learning framework.

## 5.2 Comparison between the Different Aggregation Methods Considered

Our experiments investigate various combinations of global update methods (EAA, RKLB, WB) and personalization techniques (RKLB, WB). Statistical tests, presented in Appendix B, support our observation that these configurations yield comparable performance. To reduce redundancy and enhance readability, we therefore restrict the results reported in the subsequent experiments to the WB method for both global updates and personalization.

### 5.3 Effect of $\lambda$ on the Trade-off between Performance on Local and Global Data

We analyze the impact of the personalization parameter $\lambda$ on model performance across three datasets under heterogeneity simulated using the Dirichlet distribution. Across all datasets, we observe a consistent trade-off between performance on global and local data. Results on CIFAR-10 are reported in Figure 2 and the results on FashionMNIST and SVHN are reported in Appendix C. The global data, i.e., the union of all client test sets, are approximately uniform across classes while the local data follow distinct Dirichlet distributions. Notably, $\lambda = 0$ corresponds to the global model, whereas $\lambda \to \infty$ represents the local model. As $\lambda$ increases, performance on the global distribution deteriorates in terms of accuracy, calibration (ECE), and uncertainty quantification (measured by NLL), whereas local performance improves and remains stable within a certain range of $\lambda$. The increased ECE and NLL for local models ($\lambda \to \infty$) on global data suggest that over-personalization may reduce generalization and model confidence, particularly for underrepresented classes. Conversely, at lower values of $\lambda$, the model achieves improved global performance but fails to effectively capture client-specific distributions, showing reduced confidence on local data. These results underscore the importance of $\lambda$ in controlling the trade-off between generalization and personalization, allowing the model to adapt effectively to heterogeneous data distributions in non-i.i.d. federated settings.

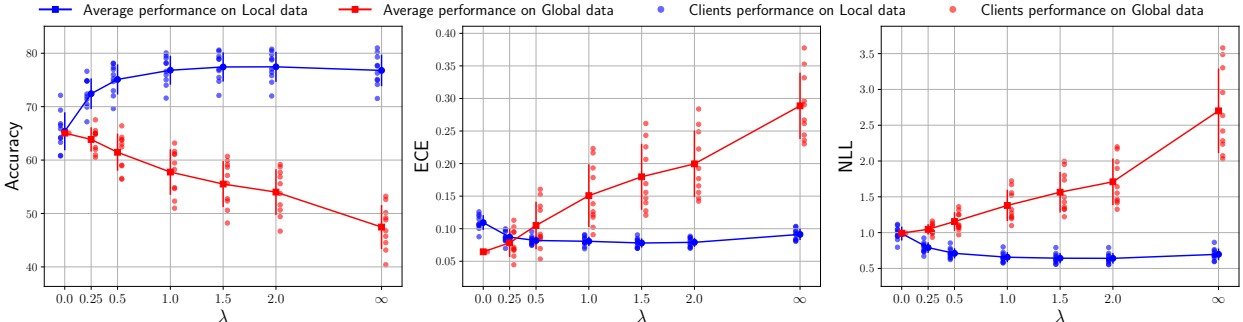

Figure 2: Effect of $\lambda$ on performance across local and global data distributions. Results are reported for the CIFAR-10 dataset. Notably, $\lambda = 0$ corresponds to the global model, whereas $\lambda \to \infty$ corresponds to the local model.

### 5.4 Comparison with State-of-the-Art

In this section, we provide a detailed analysis of the results presented in Table 2 and Figure 3. All reported values correspond to the mean and standard deviation computed over three independent runs.

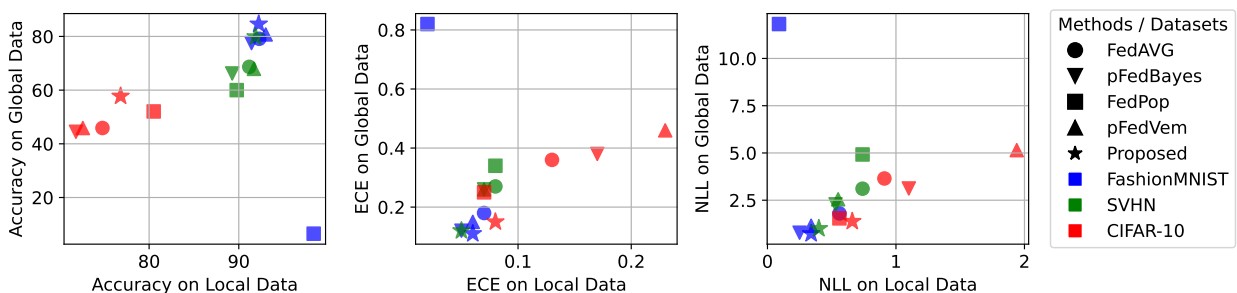

Figure 3: Trade-offs between local and global performance for personalized models. Each subplot presents results for a different evaluation metric: Accuracy (left), ECE (center), and NLL (right). Points represent method–dataset pairs. For Accuracy, the top-right region indicates a better performance trade-off, whereas for ECE and NLL, the bottom-left region is preferable. Our method (with $\lambda = 1$) consistently achieves a favorable balance across all metrics and datasets.

Table 2: Comparison of all methods across four evaluation settings: personalized models on local data (PM on LD), personalized models on global data (PM on GD), global models on local data (GM on LD), and global models on global data (GM on GD). Results are reported for three datasets (FashionMNIST, SVHN, and CIFAR-10) using Accuracy, NLL, and ECE as evaluation metrics.

| Setting | Method | FashionMNIST | | | SVHN | | | CIFAR-10 | | |
|---|---|---|---|---|---|---|---|---|---|---|
| | | Acc | ECE | NLL | Acc | ECE | NLL | Acc | ECE | NLL |
| PM on LD | FedAvg | 92.28 ± 3.16 | 0.07 ± 0.03 | 0.56 ± 0.26 | 91.17 ± 3.38 | 0.08 ± 0.03 | 0.74 ± 0.29 | 74.80 ± 4.89 | 0.13 ± 0.02 | 0.91 ± 0.16 |
| | pFedBayes | 91.42 ± 3.35 | 0.05 ± 0.01 | 0.25 ± 0.09 | 89.27 ± 3.27 | 0.07 ± 0.02 | 0.53 ± 0.19 | 71.84 ± 6.25 | 0.17 ± 0.04 | 1.10 ± 0.25 |
| | FedPop | **98.35 ± 0.37** | **0.02 ± 0.004** | **0.09 ± 0.02** | 89.77 ± 3.49 | 0.08 ± 0.03 | 0.74 ± 0.28 | **80.51 ± 4.66** | **0.07 ± 0.01** | **0.56 ± 0.12** |
| | pFedVem | 93.00 ± 2.80 | 0.06 ± 0.02 | 0.34 ± 0.14 | 91.69 ± 3.03 | 0.07 ± 0.02 | 0.55 ± 0.20 | 72.61 ± 5.09 | 0.23 ± 0.04 | 1.94 ± 0.42 |
| | Ours ($\lambda = 1$) | 92.22 ± 3.23 | 0.06 ± 0.02 | 0.34 ± 0.14 | **91.87 ± 3.03** | **0.05 ± 0.02** | **0.40 ± 0.14** | 76.82 ± 4.01 | 0.08 ± 0.01 | 0.66 ± 0.10 |
| PM on GD | FedAvg | 79.12 ± 5.56 | 0.18 ± 0.05 | 1.80 ± 0.68 | 68.67 ± 8.54 | 0.27 ± 0.08 | 3.11 ± 1.13 | 45.89 ± 6.78 | 0.36 ± 0.07 | 3.65 ± 1.24 |
| | pFedBayes | 77.51 ± 5.11 | 0.12 ± 0.05 | 0.78 ± 0.25 | 66.20 ± 10.71 | 0.26 ± 0.10 | 2.28 ± 1.04 | 44.48 ± 6.50 | 0.38 ± 0.08 | 3.12 ± 0.90 |
| | FedPop | 6.53 ± 0.49 | 0.82 ± 0.02 | 11.82 ± 0.78 | 60.00 ± 11.71 | 0.34 ± 0.12 | 4.92 ± 2.28 | 52.05 ± 8.00 | 0.25 ± 0.08 | 2.86 ± 1.54 |
| | pFedVem | 80.74 ± 5.17 | 0.15 ± 0.05 | 1.15 ± 0.46 | 67.92 ± 10.31 | 0.26 ± 0.10 | 2.55 ± 1.10 | 45.88 ± 6.05 | 0.46 ± 0.06 | 5.15 ± 1.22 |
| | Ours ($\lambda = 1$) | **84.61 ± 3.41** | **0.11 ± 0.03** | **0.73 ± 0.18** | **79.87 ± 5.21** | **0.12 ± 0.04** | **1.00 ± 0.31** | **57.75 ± 6.98** | **0.15 ± 0.07** | **1.38 ± 0.33** |
| GM on LD | FedAvg | 87.89 ± 4.83 | 0.10 ± 0.04 | 0.73 ± 0.29 | 86.62 ± 3.97 | 0.11 ± 0.03 | 0.98 ± 0.31 | 61.24 ± 7.67 | 0.16 ± 0.05 | 1.20 ± 0.25 |
| | pFedBayes | 88.00 ± 5.10 | **0.06 ± 0.02** | **0.34 ± 0.13** | 85.76 ± 6.11 | 0.09 ± 0.04 | 0.67 ± 0.32 | 63.85 ± 5.17 | 0.17 ± 0.03 | 1.25 ± 0.21 |
| | FedPop | - | - | - | - | - | - | - | - | - |
| | pFedVem | **89.49 ± 3.95** | 0.08 ± 0.03 | 0.45 ± 0.16 | 86.15 ± 4.74 | 0.10 ± 0.03 | 0.81 ± 0.28 | 60.98 ± 4.50 | 0.30 ± 0.04 | 2.43 ± 0.35 |
| | Ours | 88.45 ± 4.88 | 0.08 ± 0.03 | 0.47 ± 0.20 | **87.18 ± 4.63** | **0.07 ± 0.03** | **0.58 ± 0.20** | **65.39 ± 6.74** | **0.11 ± 0.02** | **0.99 ± 0.18** |
| GM on GD | FedAvg | 87.88 ± 0.97 | 0.09 ± 0.01 | 0.76 ± 0.05 | 86.06 ± 0.55 | 0.11 ± 0.01 | 1.01 ± 0.07 | 61.63 ± 3.81 | 0.12 ± 0.03 | 1.18 ± 0.13 |
| | pFedBayes | 88.02 ± 0.39 | **0.03 ± 0.005** | **0.34 ± 0.02** | 86.03 ± 0.41 | 0.08 ± 0.005 | 0.66 ± 0.04 | 63.86 ± 1.58 | 0.16 ± 0.01 | 1.25 ± 0.06 |
| | FedPop | - | - | - | - | - | - | - | - | - |
| | pFedVem | **89.50 ± 0.23** | 0.07 ± 0.002 | 0.45 ± 0.02 | 86.32 ± 0.22 | 0.09 ± 0.004 | 0.80 ± 0.03 | 60.88 ± 1.44 | 0.29 ± 0.01 | 2.44 ± 0.10 |
| | Ours | 88.14 ± 0.60 | 0.07 ± 0.004 | 0.49 ± 0.02 | **86.54 ± 1.05** | **0.06 ± 0.01** | **0.60 ± 0.06** | **65.05 ± 3.57** | **0.06 ± 0.01** | **0.99 ± 0.09** |

**Results of Personalized Models.** It is worth noting that all methods in the comparison, except FedAvg, employ personalized models. Consequently, we compare the local models of FedAvg, i.e., the models sent by the clients before aggregation, with the personalized models produced by the other methods, including ours. The results are presented in the first two groups of rows in Table 2 and Figure 3.

- *On Local Data:* Personalized models are designed to adapt to client-specific distributions, which is clearly reflected in their local accuracy scores, shown in the first group of rows in Table 2. Our method achieves performance comparable to or better than the baselines; for example, it attains the highest accuracy on SVHN and the second-highest on CIFAR-10. In terms of calibration (ECE) and uncertainty quantification (NLL), our method consistently yields low error rates on local data, achieving the best performance on SVHN and second-best on CIFAR-10.

- *On Global Data:* A key limitation of many personalized methods is the degradation in global performance due to overfitting to client-specific distributions. This trade-off is particularly evident in FedPop on FashionMNIST, which achieves the best local accuracy but performs poorly on global evaluations. In contrast, our method maintains strong generalization, achieving the highest global accuracy across all datasets: 84.61% on FashionMNIST, 79.87% on SVHN, and 57.75% on CIFAR-10. These represent improvements of 3.87%, 11.20%, and 5.70%, respectively, over the second-best method on each dataset. Furthermore, our method consistently achieves the lowest ECE and NLL on global data, demonstrating both well-calibrated predictions and effective uncertainty quantification.

- *Trade-off Analysis:* Figure 3 visualizes the trade-off between local and global performance for personalized models across the three metrics. For accuracy, the top-right region indicates favorable performance; for ECE and NLL, better performance lies in the bottom-left region. Our method consistently appears closest to the optimal region in all three plots, demonstrating strong local accuracy without sacrificing global generalization. The plots confirm that our method achieves a superior balance across all datasets, combining high accuracy with well-calibrated and confident predictions.

**Results of Global Models.** We compare the performance of global models across all methods, excluding FedPop, which by design does not maintain a global model. The results discussed below correspond to the last two groups of rows in Table 2. The results for global models on both local and global data are closely aligned, as the GM-on-LD setting reports the average performance of the global model evaluated across all clients' local data. Our method performs strongly in the global setting, achieving the best results on SVHN and CIFAR-10 across all three metrics: accuracy, ECE, and NLL. This demonstrates not only high predictive performance but also well-calibrated and confident uncertainty estimates, highlighting the robustness of our approach even in non-personalized settings.

## 5.5 Client-Level Fairness Analysis

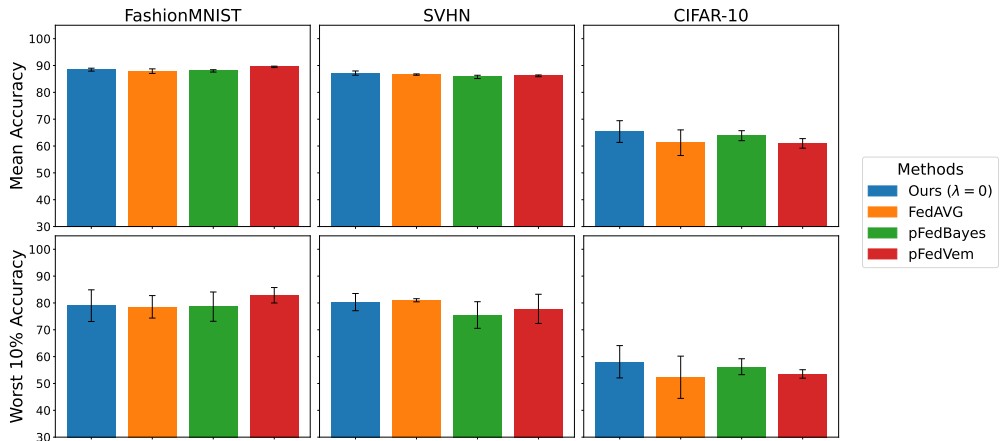

Figure 4: Performance of the global model on clients' local data. *Top row*: average accuracy. *Bottom row*: worst-case performance. Each column corresponds to a different dataset.

Despite the growing interest in BFL, its fairness implications remain a largely overlooked aspect. Although not the main focus of this work, we take a first step toward a systematic investigation of client-level fairness from an accuracy-parity perspective. In Appendix D.1, we provide an extended literature review that motivates our choice of fairness metrics.

As a comprehensive fairness evaluation requires comparing shared models across clients under consistent conditions, we focus on the performance of the global model. It is important to note that FedPop does not produce a global model. Although it achieves strong performance in terms of personalized models, the absence of a single shared model renders it unsuitable for evaluating fairness from a global perspective. As a result, the notion of fairness across all clients is not applicable in its context.

Our results, presented in Figure 4, show that the global model produced by our method is comparable to state-of-the-art approaches in terms of both mean accuracy and worst 10% accuracy across clients. Additionally, we conduct a study to examine the effect of Bayesian layers on algorithmic fairness in federated learning, and report our findings in Appendix D.2.

## 6 Broader Applicability of the Proposed Method

Although the use of information geometry in BFL is natural, our method also applies to any optimization problem defined over probability spaces rather than point estimates. The approach can naturally extend to domain adaptation scenarios: when a model is trained on source data $A$ and new target data $B$ become available, retraining only on $B$ may lead to catastrophic forgetting of the source domain. Our projection framework provides a principled trade-off, retaining prior knowledge while integrating new information. This flexibility is further enhanced by the use of IVON, which combines the efficiency of deterministic learning with the ability to perform inference and adaptation in the posterior space. However, it is important to note that our approach fundamentally differs from standard domain adaptation methods based on optimal

transport Courty et al. (2016), which operate on data distributions rather than model posteriors. Appendix E provides additional insights and discussion.

## 7 Conclusions

In this work, we proposed a personalized Bayesian Federated Learning method that balances local adaptation and global generalization while explicitly accounting for uncertainty quantification and model calibration. Our approach combines variational learning with information-geometric tools, namely projection and barycenters, to produce client-specific models that remain robust under data heterogeneity with minimal additional cost. Through experiments on three benchmark datasets, FashionMNIST, SVHN, and CIFAR-10, we demonstrate that our method consistently achieves a favorable trade-off between performance on local and global data distributions. In global evaluations, our personalized models outperform existing baselines in terms of accuracy, calibration, and uncertainty quantification, while maintaining highly competitive performance on local data. This contrasts with other methods, which often tend to overfit or underfit across clients. Furthermore, we show that the global model produced by our approach generalizes better than those of state-of-the-art alternatives. These empirical results highlight the effectiveness of principled, information-geometric personalization combined with variational learning in federated settings. Future work will explore adaptive mechanisms to control the degree of personalization, extend our personalization framework to non-parametric BFL, and evaluate its performance under additional heterogeneity scenarios, such as feature and quantity shifts, as well as on real-world cross-device FL benchmarks with naturally occurring client partitions.

## Acknowledgments

The work has been supported by the EC through the Horizon Europe/JU SNS project, ROBUST-6G (Grant Agreement no. 101139068), the IMT "Futur, Ruptures & Impacts" program, and the European Research Council (ERC) under the European Union's Horizon 2020 Research and Innovation programme (Grant agreement No.101003431).

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

# A   Proof of Theorem 1

*Proof.* In our proof, we directly address the problem through the KKT conditions, leveraging the well-known variational structure of projection problems (see Eq. 5.14, Ekeland & Temam (1999)). This approach allows us to highlight the key concepts and ideas underlying the proof without unnecessary technical overhead. All cited Theorems refer to Ekeland & Temam (1999).

**Weak or Strong Duality:** We first apply Proposition 5.1 (generalization of Slater's condition) to show that the projection problem is stable. Subsequently, Proposition 2.2 then implies that it is also normal. Finally, Lemma 5.2 verifies the hypotheses of Proposition 2.1, which yields zero duality gap. Consequently, the max-min and min-max formulations attain the same finite optimal value, establishing strong duality.

**Relations between the min-max and max-min solutions:** Theorem 5.1 ensures that, for fixed Lagrangian multipliers, if one can identify a primal variable that satisfies the saddle-point condition, then this variable solves the primal problem for an appropriate constraint value. As shown in the proof, this is equivalent to minimizing the Lagrangian with respect to the primal variable while keeping the multipliers fixed. This guarantees that the solution obtained via the barycenter coincides with the solution of the original projection problem.

We restate the problem for clarity. The projection problem is defined as:

$$\min_{p \in \mathcal{M}} D(p|p_g) \quad \text{s.t.} \quad D(p|p_k) \le r_k. \tag{11}$$

Assumption 2 ensures that $D(\cdot|p_g)$ and $D(\cdot|p_k)$ are convex in their first argument. Consequently, the problem is a convex optimization problem with a convex inequality constraint.

Because $p_k \in \mathcal{S}_k$ and the interior of $\mathcal{S}_k$ is nonempty (e.g., any $q$ sufficiently close to $p_k$ lies in the interior), the constraint admits a Slater point. Therefore, the KKT conditions are both necessary and sufficient for optimality.

The Lagrangian for problem 11 is:

$$\mathcal{L}(p, \lambda) = D(p|p_g) + \lambda(D(p|p_k) - r_k), \qquad \lambda \ge 0. \tag{12}$$

Let $p^\star = p_{g,k}$ be the minimizer of 11. The KKT conditions are:

1. Primal feasability: $D(p^\star|p_k) \le r_k$.

2. Dual feasibility: $\lambda^\star \ge 0$.

3. Complementary slackness:
$$\lambda^\star\big(D(p^\star|p_k) - r_k\big) = 0. \tag{13}$$
   Since varying the radius $r_k$ parametrizes the geodesic between $p_g$ and $p_k$, and $p_g \notin \mathcal{S}_k$ in any nontrivial personalization setting, the solution necessarily lies on the boundary, i.e., $D(p^\star|p_k) = r_k$, which implies $\lambda^\star > 0$.

4. Stationarity:
$$\nabla_p D(p^\star|p_g) + \lambda^\star \nabla_p D(p^\star|p_k) = 0. \tag{14}$$

Divide equation 14 by $(1 + \lambda^\star)$:

$$\frac{1}{1 + \lambda^\star}\nabla_p D(p^\star|p_g) + \frac{\lambda^\star}{1 + \lambda^\star}\nabla_p D(p^\star|p_k) = 0. \tag{15}$$

Define normalized nonnegative weights

$$w_g = \frac{1}{1 + \lambda^\star}, \qquad w_k = \frac{\lambda^\star}{1 + \lambda^\star}. \tag{16}$$

Then equation 14 becomes:

$$w_g \nabla_p D(p^\star | p_g) + w_k \nabla_p D(p^\star | p_k) = 0. \tag{17}$$

The barycenter of $\{p_g, p_k\}$ with weights $w_g, w_k$ is defined as the minimizer of:

$$\min_{p \in \mathcal{M}} \left( w_g D(p|p_g) + w_k D(p|p_k) \right). \tag{5}$$

Since the objective is convex, its minimizer must satisfy the first-order optimality condition:

$$w_g \nabla_p D(p^\star | p_g) + w_k \nabla_p D(p^\star | p_k) = 0. \tag{18}$$

This is exactly equation 17, obtained from the KKT stationarity condition. Thus, the minimizer of the constrained projection problem 11 coincides with the barycenter:

$$p^\star = p_D^*(\{p_g, p_k\}, \{w_g, w_k\}). \tag{19}$$

This concludes the proof. $\qquad\square$

# B Comparison between the Different Aggregation Methods Considered

We conduct experiments with various combinations of global update and personalization methods, including EAA, RKLB, and WB for global aggregation, and RKLB and WB for personalization. Across all configurations, we observe consistent trends and comparable results. To confirm these observations, we apply the Wilcoxon signed-rank test Demšar (2006) to compare the performance of different aggregation methods across multiple runs (random seeds) and datasets. We compute pairwise Wilcoxon tests between all methods to assess whether their performance differences are statistically significant. This yields a matrix of $p$-values indicating, for each pair of methods, whether one consistently outperforms the other. The results of the Wilcoxon signed-rank tests, presented in Figure 5, show that in most cases the $p$-values are high, indicating no statistically significant difference between the aggregation methods. This suggests that, despite small fluctuations in performance across datasets, the aggregation methods perform similarly overall, and no single aggregation approach consistently outperforms the others in a statistically meaningful way across the considered metrics. To further illustrate the variability and stability of the aggregation methods across different datasets, we provide violin plots in Figure 6. These plots show the distribution of global model accuracies across multiple random seeds for each aggregation method on FashionMNIST, SVHN, and CIFAR-10. The shape and spread of the distributions offer insight into both the typical performance (median) and variability (spread) of each method. Based on these results, which show that the WB-based aggregation achieves competitive and consistently stable performance across datasets, we restrict the experiments reported in the main text to this method for both the global update and personalization steps.

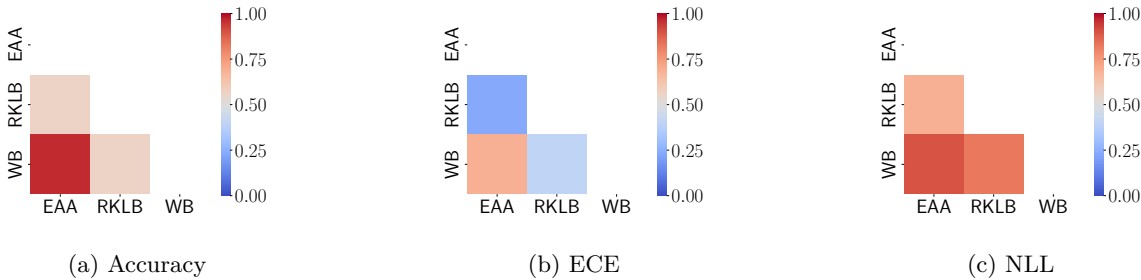

(a) Accuracy        (b) ECE        (c) NLL

Figure 5: Wilcoxon signed-rank test $p$-values comparing aggregation methods across all datasets for three evaluation metrics: (a) accuracy, (b) ECE, and (c) NLL. Lower $p$-values indicate statistically significant differences between methods. Only the lower triangle of each matrix is shown to avoid redundancy.

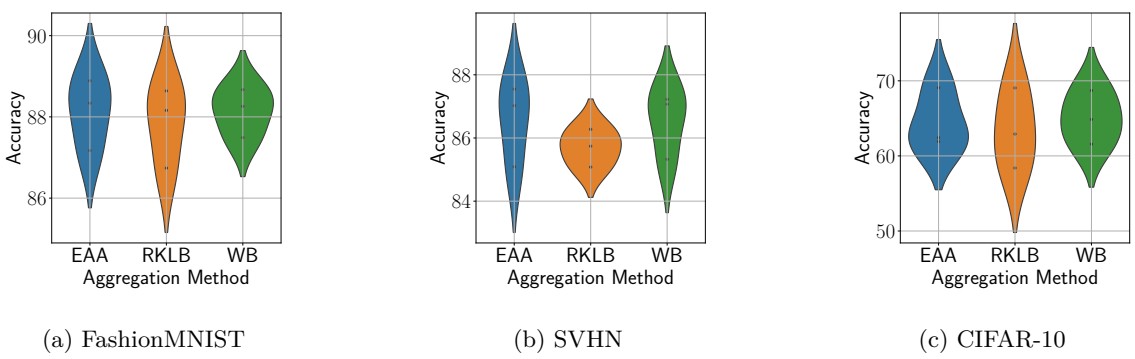

(a) FashionMNIST        (b) SVHN        (c) CIFAR-10

Figure 6: Accuracy distributions across random seeds for different aggregation methods on: (a) FashionM-NIST, (b) SVHN, and (c) CIFAR-10 datasets. A lower spread and higher median indicate better and more stable performance.

## C   Effect of $\lambda$ on the Trade-off between Performance on Local and Global Data

To complement Section 5.3 in the main text, we report in Figures 7 and 8 the results illustrating the effect of $\lambda$ on the trade-off between performance on local and global data for the FashionMNIST and SVHN datasets.

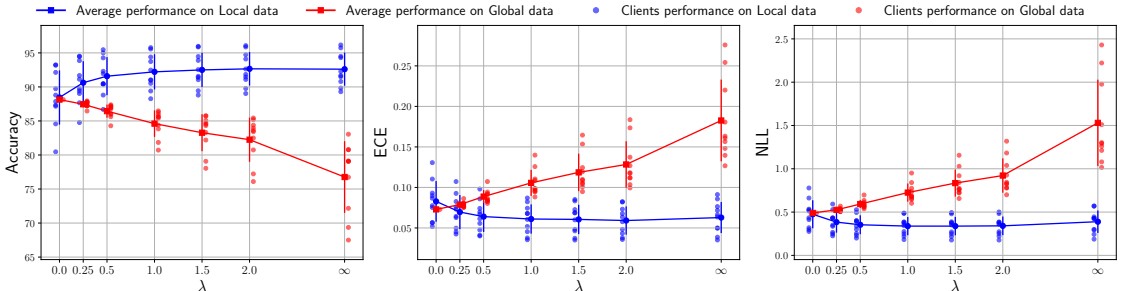

Figure 7: Effect of $\lambda$ on the trade-off between performance on local and global data distributions. Results are reported for the FashionMNIST dataset. Notably, $\lambda = 0$ corresponds to the global model, whereas $\lambda \to \infty$ corresponds to the local model.

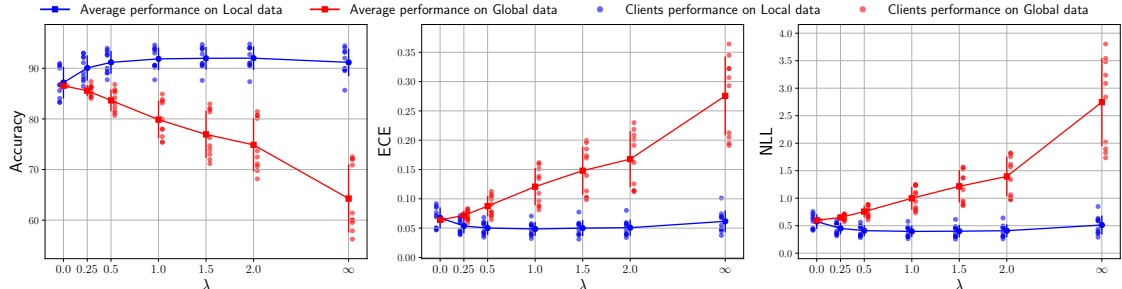

Figure 8: Effect of $\lambda$ on the trade-off between performance on local and global data distributions. Results are reported for the SVHN dataset. Notably, $\lambda = 0$ corresponds to the global model, whereas $\lambda \to \infty$ corresponds to the local model.

# D   Client-Level Fairness Analysis

Client heterogeneity introduces significant challenges in distributed learning architectures, one of which is ensuring algorithmic fairness. Algorithmic fairness refers to the impartial treatment of individuals or groups in automated decision-making processes, without bias, discrimination, or favoritism based on innate or acquired characteristics Saxena et al. (2019); Mehrabi et al. (2021). In the context of FL, where models are collaboratively trained on data from multiple sources representing diverse populations, overlooking fairness considerations can amplify or perpetuate existing societal biases in the global model, leading to unfair outcomes for certain subgroups. To mitigate this issue, a growing body of research has proposed fairness-aware FL algorithms, including approaches based on personalized FL Tan et al. (2023); Jeong & Kountouris (2023), which tailor models to individual client data distributions. Shi et al. (2023) presents a comprehensive taxonomy of fairness-aware FL strategies, covering key aspects such as client selection, optimization, contribution assessment, and incentive allocation.

## D.1   Related Work

**Fairness in Federated Learning.**   Fairness is a multifaceted concept that spans several disciplines, including the social sciences, law, machine learning, and statistics, each offering distinct perspectives and implicitly different definitions. In the context of machine learning, Mehrabi et al. (2021) provides a comprehensive overview of key fairness notions, categorizing them into three primary types: *individual fairness*, which seeks to ensure similar outcomes for similar individuals; *group fairness*, which focuses on equal treatment across predefined demographic groups; and *subgroup fairness*, which combines elements of both individual- and group-based approaches to better capture fairness across a broader range of population segments.

In FL, fairness extends beyond the behavior of a single predictive model to encompass the equitable treatment of clients participating in the distributed training process. Because clients contribute heterogeneous data and resources, an FL system is considered fair if it avoids systematically privileging or disadvantaging certain participants. To formalize fairness in this context, we draw on two long-standing philosophical traditions, **utilitarianism** and **egalitarianism**, which have recently been adapted to the federated setting Zhang et al. (2022a); Hu et al. (2022b); Wang et al. (2021); Gao et al. (2024), yet remain underexplored within the BFL paradigm.

- **Utilitarianism**, rooted in the works of Bentham and Mill Bentham (1890); Bentham & Mill (2004) and formalized by Maskin Maskin (1978), evaluates the fairness of a system through the aggregate welfare it produces. In FL, this corresponds to optimizing global utility, for example, the average or mean accuracy across clients.

- **Egalitarianism**, inspired by Rawls' *difference principle* Rawls (1974; 1999), instead focuses on protecting the worst-off participants. Translated to FL, this notion aligns with max–min fairness, which aims to maximize the minimum (worst-case) performance across clients, typically evaluated through the worst-10% accuracy metric.

**Wasserstein Barycenters for Fairness.**   In the domain of fair learning, recent work has highlighted the effectiveness of WBs as a unifying framework for promoting fairness in ML while preserving strong predictive performance. Chzhen et al. (2020) investigated fair regression by establishing a direct connection between the demographic parity constraint and the WB problem. They showed that the optimal fair predictor, minimizing the squared error while ensuring independence from sensitive attributes, can be derived as the WB of the conditional distributions corresponding to different sensitive groups. This approach leads to a simple yet powerful post-processing technique that transforms any regression model into a fair one without requiring retraining. Importantly, their method provides robust, distribution-free fairness guarantees, improving fairness metrics with only minimal reductions in predictive accuracy.

Building on this foundation, Gaucher et al. (2023) extended the use of WBs to classification tasks. They demonstrated that demographic parity in classification can be achieved by solving a fair regression problem, followed by appropriate thresholding. Their approach underscores the role of the WB in aligning group-wise distributions, thereby reducing disparities across sensitive attributes. In settings with binary sensitive

attributes, the barycenter plays a central role in determining optimal classification thresholds, enabling a favorable trade-off between fairness and performance. Together, these works provide both theoretical justification and practical algorithms for transport-based fairness, offering a cohesive framework applicable to both regression and classification.

Our work connects with these findings by incorporating WBs during the aggregation phase. We extend this line of research by analyzing WBs through the lens of client-level fairness definitions, offering a complementary perspective to the discussions presented in this section.

### D.2 Effect of Variational Inference Layers on Client-Level Fairness

In this subsection, we investigate how increasing the number of variational inference layers (i.e., Bayesian layers implemented via variational inference) in the model architecture affects fairness outcomes. To this end, we focus on the BA-BFL setting implemented within the Hybrid Bayesian Deep Learning framework Jamoussi et al. (2024), alongside the standard FedAvg baseline.

It is important to note that both the WB and RKLB aggregation strategies reduce to simple arithmetic averaging in the deterministic limit (i.e., when posterior variances vanish). Consequently, BA-BFL with zero Bayesian layers is equivalent to FedAvg, enabling a unified comparison in Figure 9. To ensure a fair comparison, we restrict our analysis to BA-BFL and FedAvg, deliberately excluding additional mechanisms specific to other Bayesian methods, including our IVON-based approach, since the differing optimizers prevent a direct comparison with FedAvg in this context. This controlled setup allows for a focused examination of how the degree of Bayesian modeling, quantified by the number of variational inference layers, influences the fairness of the global model across different clients. However, our analysis does not reveal a consistent trend across experiments, suggesting that the relationship between the number of Bayesian layers and client-level fairness may depend on other factors, such as task complexity.

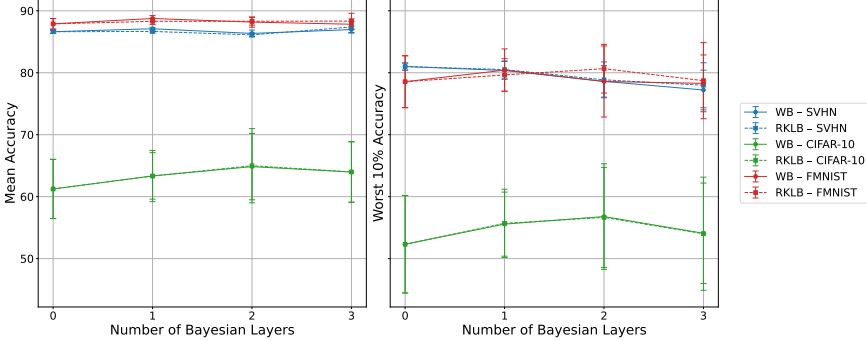

Figure 9: Impact of Variational Inference Layers on Fairness.

# E   Beyond BFL: Broader Applicability of the Proposed Method

## E.1   Merging Variational Foundation Models

Foundation Models (FMs) have emerged as powerful general-purpose learners, capable of adapting to a wide range of downstream tasks after large-scale pretraining. However, as data distributions shift and new domains emerge, keeping these models up to date without retraining from scratch remains a major challenge Bommasani & et al. (2021). Continual pretraining Ke et al. (2023); Shi et al. (2024), fine-tuning Hu et al. (2022a), and model merging McMahan et al. (2017); Matena & Raffel (2022); Daheim et al. (2023); Goddard et al. (2024) offer promising paths forward, enabling FMs to integrate new knowledge while retaining broad generalization. However, updating such large systems at scale faces key challenges: computational constraints, catastrophic forgetting McCloskey & Cohen (1989), and potential misalignment in uncertainty quantification. Addressing these challenges calls for principled and efficient update rules that incorporate domain-specific adaptations into a global FM while preserving statistical rigor and scalability.

Within the spectrum of adaptation strategies, variational approaches provide a rigorous framework for representing model uncertainty, making them particularly suitable in contexts where both reliability and interpretability are critical. We focus on variational FMs whose parameters encode posterior distributions, enabling updates to be expressed as operations on the statistical manifold of distributions. We assume these models are pretrained or fine-tuned using the IVON optimizer Shen et al. (2024), chosen for its ability to match Adam's computational cost while delivering strong Bayesian performance in large-scale settings. IVON has been shown to pretrain GPT-2 on OpenWebText and ResNet on ImageNet from scratch, as well as to fine-tune large masked language models (e.g., DeBERTaV3). Within this setting, we formulate FM merging as an information-geometric projection from a global model, i.e., the pretrained model, onto a sphere centered at a specialized model, i.e, a fine-tuned model. This formulation naturally extends our earlier notion of specialization, which we showed to be equivalent to computing a barycenter in the variational parameter space, toward multi-model aggregation via barycentric averaging that minimizes the average information-geometric discrepancy across multiple fine-tuned models. Our approach thus provides an interpretable and theoretically grounded merging mechanism that generalizes existing techniques such as Fisher-weighted averaging Matena & Raffel (2022), as well as mixture and product-of-experts methods.

**Related Work: Model Merging.**   Originally proposed in the context of FL to mitigate communication overhead and enhance privacy McMahan et al. (2017), model merging has since been adopted in both computer vision and large language models Goddard et al. (2024). Wortsman et al. (2022) demonstrated that averaging the weights of models fine-tuned under varied hyperparameters can improve both accuracy and out-of-distribution robustness. Using only a few fine-tuned models, Stock Jang et al. (2024) achieved robust merges via layer-wise linear interpolations that explicitly operate in the Euclidean geometry of the parameter space. In contrast, our method focuses on the manifold geometry of the variational posteriors.

**Generalization via $D$-Barycenters.**   Let $\{p_k\}_{k=1}^N$ denote the variational posteriors (e.g., IVON-trained posteriors of fine-tuned FMs), and let $p_k(y|x)$ denote the predictive distribution of the $k^{\text{th}}$ FM.

- **Forward KL.** With $D = \text{KL}(p_k\|q)$, the minimizer is the *mixture* in posterior space: $p_D^\star(\theta) = \sum_{k=1}^N w_k\, p_k(\theta)$. After marginalizing over $\theta$, the resulting predictive distribution also mixes pointwise as $p_D^\star(y \mid x) = \sum_{k=1}^N w_k\, p_k(y \mid x)$, a construction commonly referred to as the *Mixture of Experts*.

- **Reverse KL.** With $D = \text{KL}(q\|p_k)$, the solution is the *log-opinion pool* or *Product of Experts*: $p_D^\star(\theta) \propto \prod_{k=1}^N p_k(\theta)^{w_k}$. In exponential families, this corresponds to *natural-parameter averaging*. For Gaussians posteriors, $\Lambda^\star = \sum_{k=1}^N w_k\, \Lambda_k$ and $\mu^\star = (\Lambda^\star)^{-1} \sum_{k=1}^N w_k\, \Lambda_k \mu_k$, where $\Lambda_k$ denotes the precision matrix of the $k^{\text{th}}$ model. This formulation connects directly to Fisher merging Matena & Raffel (2022).

- **Wasserstein-2.** With $D = \text{W}_2^2(p_k\|q)$, the minimizer is the *Wasserstein-2 barycenter*. For Gaussians distributions, the barycenter remains Gaussian with $\Sigma_{\text{W}_2^2} = \left(\sum_{k=1}^N w_k \Sigma_k^{\frac{1}{2}}\right)^2$, $\mu_{\text{W}_2^2} = \sum_{k=1}^N w_k \mu_k$ often yielding more robust summaries than naive parameter averaging.

**Practical implications.** Barycentric merging provides a single, interpretable control (the weights $\{w_k\}$) to balance global and domain-specific knowledge. It recovers popular FM-merging schemes as special cases and admits closed-form solutions for common variational families (e.g., diagonal Gaussians) under widely used divergences. When complemented by the IVON training regime, the weights can be instantiated from curvature estimates, such that higher-curvature models receive greater weight Daheim et al. (2023); Shen et al. (2024). Consequently, high-uncertainty (low-curvature) models are down-weighted during aggregation.

**Limitations and Possible Extensions.** Like most merging methods in distribution space (Bayesian) or parameter space (deterministic), our approach assumes architecturally aligned models, i.e., compatible layers and widths, to enable layer-wise aggregation. This constraint is particularly limiting for FMs, where specialized models are often smaller than the pretrained backbone. As a next step, we aim to relax this assumption through Gromov–Wasserstein optimal transport maps (e.g., see Delon et al. (2022); Le et al. (2022)), which enable mappings between spaces of different dimensionalities. Furthermore, we plan to conduct FM-scale experiments to assess the scalability and efficiency of the method in large-scale settings.

### E.2 Incremental Learning

We design a toy example to simulate a continual learning scenario involving two sequential tasks: Task A, consisting of the first five classes of MNIST, and Task B, corresponding to the remaining five. We first train a model (Model A) on Task A. Subsequently, Task B becomes available, while access to Task A is revoked, mimicking a typical continual learning setup where previously seen data are no longer accessible due to constraints such as data retention policies or regulations like the General Data Protection Regulation (GDPR). We evaluate multiple strategies: training a new model (Model B) solely on Task B, fine-tuning Model A on Task B with and without Elastic Weight Consolidation (EWC) Kirkpatrick et al. (2017), and computing the barycenter of Model A and Model B. As shown in Figure 10, the barycentric combinations offer a favorable trade-off between performance on both tasks, preserving accuracy on Task A while adapting effectively to Task B, thereby mitigating the catastrophic forgetting observed in the other scenarios.

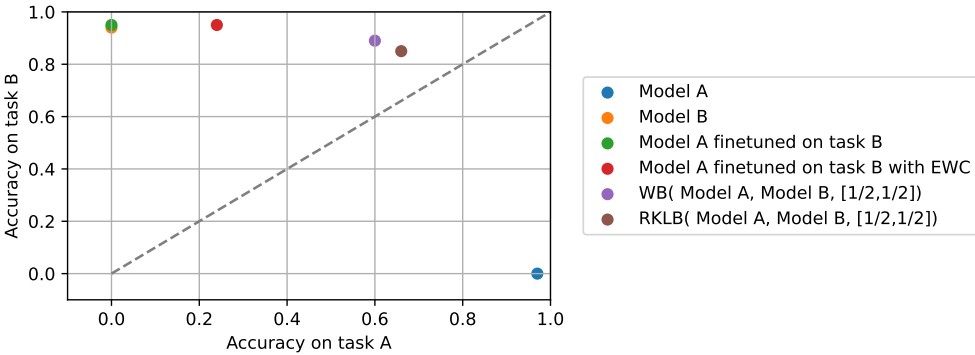

Figure 10: Accuracy Trade-off in an Incremental Learning Setting

## F    Additional Discussion on Barycenters

**Wasserstein Barycenters in Machine Learning.** The Wasserstein Barycenter (WB) has emerged as a powerful tool in a wide range of machine learning applications, including Bayesian learning, multimodal representation learning, and fair learning.

The Bayesian Wasserstein Barycenter (BWB), introduced in Backhoff-Veraguas et al. (2022), minimizes the Wasserstein Bayes risk corresponding to the $p$-Wasserstein distance as the loss, yielding a predictive posterior distribution with lower variance compared to Bayesian Model Averaging (BMA), i.e., the Bayes estimator that minimizes the Bayes risk under the $L_2$ distance or the KL divergence. In addition, the BWB enhances interpretability by respecting the geometric structure of the model space. It extends naturally to nonparametric model spaces and can be computed efficiently using stochastic gradient descent.

In the context of multimodal learning, Qiu et al. (2024) redefines the aggregation of unimodal inference distributions in Variational Autoencoders (VAEs) using the WB. Here, the WB is formulated as the central distribution that minimizes the average discrepancy in terms of the squared 2-Wasserstein distance:

$$\nu^\star = \arg\min_{\nu \in \mathcal{P}(\mathcal{X})} \sum_{k=1}^{N} w_k W_2^2(\mu_k, \nu), \quad \sum_{k=1}^{N} w_k = 1,$$

with $W_2$ denotes the 2-Wasserstein distance. Unlike traditional aggregation methods such as the Product of Experts or Mixture of Experts, which rely on the asymmetric KL divergence, the WB better preserves the geometric structure of distributions and enables smooth interpolation across modalities. In multimodal VAEs, WB-based models such as WB-VAE and its variant MWB-VAE achieve superior classification accuracy and conditional generation performance, particularly on datasets with missing modalities.

The works in Backhoff-Veraguas et al. (2022); Qiu et al. (2024) collectively highlight the flexibility and scalability of WB-based methods. Their results demonstrate the WB's potential to unify diverse distributions while maintaining geometric interpretability and robustness.

**Forward KL Barycenter.** As noted in Jamoussi et al. (2024), the Forward Kullback-Leibler (FKL) barycenter of Gaussian distributions, one of the $\alpha$-divergence barycenters is not itself Gaussian. While it is possible to approximate the FKL barycenter with a Gaussian distribution through projection onto the space of Gaussians D'Ortenzio et al. (2022), this approach fails to preserve parameter independence, as shown in Figure 11, making it a suboptimal choice for practical implementation. The following equations characterize the parameters of the Gaussian approximation to the FKL Barycenter for a set of $N$ Gaussian distributions with parameters $\{(\mu_k, \Sigma_k)\}_{k=1}^{N}$.

$$\Sigma_{\text{FKL}} = \sum_{k=1}^{N} w_k \left( \Sigma_k + (\mu_k - \mu_{\text{FKL}})(\mu_k - \mu_{\text{FKL}})^T \right), \qquad \mu_{\text{FKL}} = \sum_{k=1}^{N} w_k \mu_k.$$

Figure 11: Example of barycenters between two multivariate Gaussian distributions with independent parameters.

