# OpenReview forum: "Cost-Free Personalization via Information-Geometric Projection in Bayesian Federated Learning"
_TMLR — Accepted by TMLR_

### Review · Reviewer_huuH · 2025-11-22

**Summary Of Contributions:**

This paper introduces a novel personalization technique for Bayesian Federated Learning (BFL) utilizing information-geometric projection. The core concept involves projecting the global posterior onto a divergence-defined sphere encompassing a client’s local posterior, thereby enabling a customizable trade-off between global generalization and local specialization. Under mild convexity assumptions, the authors establish that this projection is equivalent to computing a two-point barycenter, resulting in a closed-form solution. This approach eliminates the need for additional fine-tuning, computation, or data access, making it “cost-free” personalization. The method is evaluated within a variational learning framework using the IVON optimizer and compared against several BFL baselines across three vision datasets. The results demonstrate favorable trade-offs in both global and local performance.

**Strengths**
- Frames personalization as an information-geometric projection and connects projection to barycenters via a clear theoretical result (Theorem 1).
- As demonstrated through meticulous experimental validation, the method not only competes with but frequently surpasses the existing baselines.

**Weaknesses**
- Results rely heavily on Gaussian mean-field posteriors to achieve "cost-free" personalization.
- The optimizer is highly sensitive to hyperparameters.

**Audience:**

Yes

**Audience Explanation:**

The paper sits at the intersection of:
- Bayesian deep learning
- Personalized federated learning
- Information geometry

The method presents a novel conceptual perspective, offering practical benefits and ease of interpretation.  Furthermore, the experiments evaluate common benchmarks, suggesting this work provides a valuable approach to addressing data heterogeneity.

**Broader Impact Concerns:**

There are no ethical concerns to raise regarding this work.

**Claims And Evidence:**

Yes

**Claims Explanation:**

Overall, the core claims are well-supported,
- The theoretical equivalence between projection and barycenter is clearly stated and relies solely on convexity (Assumption 2).
- Experiments do support the claim that the method improves the global/local trade-off.
- The entire experimental section is clearly justified, as it provides detailed information about the selection and construction of baselines.

**Requested Changes:**

No adjustments are required.

---

> ### Author Response · Authors · 2025-11-26
> **Authors Response**
>
> We thank the Reviewer for their constructive feedback. We would like to inform the Reviewer that we have updated the paper following the comments and the requests of Reviewer km55.

---

> ### Author Response · Authors · 2025-12-21
> **Authors Response - Visibility correction**
>
> We thank the Reviewer for their feedback. We appreciate the assessment that the method offers a novel conceptual perspective with practical benefits and ease of interpretation, and that the experiments on common benchmarks suggest a valuable approach to addressing data heterogeneity.
> We apologize that our previous response was inadvertently left with restricted visibility in the system; this has now been corrected.
> We would like to inform the Reviewer that the manuscript has been updated in response to the reviewers’ recommendations.

---

### Review · Reviewer_km55 · 2025-11-22

**Summary Of Contributions:**

The authors' work contribute to Federated Learning (FL), in particular, to Private FL (PFL). The workflow in PFL consists of 3 iterative stages:
1. A *learning phase* in which clients train privately their own local models with their own private data.
2. An *aggregation phase* in which the server aggregates the local model parameters retrieved from each client into an updated global model.
3. An *update phase* in which the server sends the updated global model parameters to each client.

The authors focus on the 3rd stage, allowing for the *personalization* of the updated global model. Roughly speaking, each client has the choice to replace their local model parameters by the updated global model parameters, keep their own local model parameters, or find a compromise between both. To achieve this, given a statistical divergence function, they propose to use a projection of the global model onto a ball centered at the local model; the choice of radius determines the trade-off between one model and the other. As this projection step could be possibly intractable, the authors show it can be recast as computing a suitable barycenter with respect to the same statistical divergence. The personalization step thus adds flexibility to the PFL workflow. To ensure that the implementation is tractable, they make the strong assumption that all models are multivariate normal distributions with diagonal covariance matrices. This allows them to use explicit expressions for the barycenters for the statistical divergences that they use.

**Additional Comments:**

No comments.

**Audience:**

Yes

**Audience Explanation:**

Private Federated Learning is a topic of interest for the community. Enabling personalization for each one of the clients is also a topic of interest, particularly as there is a trade-off in improving the performance of local models on local data at the expense of poor generalization, and possibly suboptimal global models.

**Broader Impact Concerns:**

I do not have broader impact concerns due to the nature of the contribution. However, the authors do provide a nice discussion about potential issues related to fairness in Apoendix D, particularly when overpersonalization may generate poor performance on underrepresented data.

**Claims And Evidence:**

No

**Claims Explanation:**

I find the work interesting and the evidence provided by their experiments sufficient. However, the proof of their main theoretical claim, namely, Theorem 1, seems to be lacking. The authors start from the projection problem, which is convex under Assumption 2 in the paper. They then form a Lagrangian to connect the problem of computing a projection to that of computing a barycenter. However, for this connection to be *an equivalence* the authors need to prove at least that strong duality holds. In the notation in Appendix A, one always has *weak duality*, that is,

$$\sup\_{\lambda \geq 0}\inf\_{p} \mathcal{L}\_{\lambda}(p) \leq \inf\_{p} \sup\_{\lambda \geq 0} \mathcal{L}_{\lambda}(p).$$

Computing the projection is equivalent to the problem in the right-hand side in the above inequality, whereas computing the barycenter for fixed $\lambda$ corresponds to the problem

$$\inf\_{p} \mathcal{L}\_{\lambda}(p).$$

However, it could be the case that *strong duality* does not hold, in which case

$$\sup\_{\lambda \geq 0}\inf\_{p} \mathcal{L}\_{\lambda}(p) < \inf\_{p}\sup\_{\lambda \geq 0} \mathcal{L}\_{\lambda}(p).$$

This would imply that the best choice for $\lambda$ does not match the minimal divergence obtained when computing the projection. In particular, one would *not* have an equivalence between computing a projection and computing a barycenter. This gap needs to be addressed by proving that, for the problems that the authors consider, strong duality actually holds.

Note that even if strong duality holds, the optimal distributions for the left and right-hand sides may not be the same. For this stronger conclusion *total duality* must hold.

To be clear, my contention is not with the main contribution of the paper. In fact, I think that there is no issue in proposing the computation of a barycenter between the global and local model as a method for personalization since, as the authors also discuss, this enables a trade-off between local and global models. My main issue is with Theorem 1, for which the proof is, at this stage, incomplete.

**Requested Changes:**

1. In pg. 3, the second to last paragraph, change "distributions ith respect" by "distributions with respect".
2. In Section 3.1, is the parametric distribution family $\mathcal{Q}$ the same for each client and the server? I suppose this is the case, but I suggest stating this explicitly for an audience less familiar with PFL.
3. In Section 3.2, for (6) it is stated that $h$ in the denominator is "the Hessian approximation". However, (6) shows a relation between scalars. It seems that either there is a typo, or an explanation of what $h$ is is missing.
4. Appendix A is not referenced in the main text. It should be stated that the proof of Theorem 1 is provided in Appendix A.
5. The proof of Theorem 1 must be corrected as indicated before.
6. Appendix C is not referenced in the main text.
7. Why do the authors use the parameter $\lambda$ directly to characterize the trade-off? It seems that using $\lambda/(1 + \lambda)$ or $1/(1 + \lambda)$ would be more interpretable, as these parameters take values on $[0, 1]$ instead of $[0, \infty]$, and computing the barycenters uses these weights, as discussed in Remark 3.
8. The experimental results show the effect of personalization, i.e., the value of $\lambda$, on the local models. However, I did not find a similar discussion for the global model. One would expect that in the limit of overpersonalization, $\lambda \to \infty$, the global model never gets truly updated beyond the first step.
9. As far as I understand, the authors consider the same level of personalization for each client. What is the effect of letting each client select their own personalization factor?

---

> ### Author Response · Authors · 2025-11-26
> **Authors Response**
>
> We thank the Reviewer for their constructive feedback.
>
> We would first like to clarify a misunderstanding that seems to have influenced the Reviewer’s summary as well as Comments 8 and 9. The Reviewer assumed that personalization is performed at every update step.
> Although the idea of continuous personalization along the training is interesting and can be explored as future work, in our framework personalization is performed only once, after FL training has completed. Therefore, the personalization step does not affect the global model during training. *We have added a sentence in Section 4 (line 4) to make this explicit.*
>
> Below, we respond to each requested change point by point.
>
> ### Comments on Theorem 1
>
> In our proof, we address the problem directly via the KKT conditions, leveraging the standard variational characterization (structure) of projection problems (see Eq. 5.14 in [1]). This approach allows us to emphasize the essential concepts underlying the proof, highlighting the key ideas without obscuring them with unnecessary technicalities. That said, we agree that the exposition would benefit from additional rigor and clearer justification of the steps involved. We therefore add the clarifications below to make the argument fully precise. Unless otherwise stated, all cited results refer to [1].
>
> **Weak vs. Strong Duality:** We apply Proposition 5.1 (a generalized Slater-type condition) to show that the projection problem is stable. Subsequently, Proposition 2.2 then implies that the problem is also normal. Finally, Lemma 5.2 guarantees that the hypotheses required to apply Proposition 2.1 are satisfied, which yields zero duality gap. Consequently, the max-min and min-max formulations attain the same finite (optimal) value, establishing strong duality.
>
> **Relations between the min-max and max-min solutions:** Theorem 5.1 ensures that, for fixed Lagrangian multipliers, if we can identify a primal variable that satisfies the saddle-point condition, then this variable optimally solves the primal problem for some constraint value. As we show in the proof, this is equivalent to minimizing the Lagrangian with respect to the primal variable while keeping the multipliers fixed. This guarantees that the solution computed via the barycenter coincides with the solution of the original projection problem.
>
> *To avoid any further ambiguity, we have incorporated these observations into the revised proof to provide complete context and make the logical dependencies explicit.*
>
> Source:
> [1] Convex Analysis and Variational Problems, Ivar Ekeland and Roger Temam
>
> ### Responses to Requested Changes
>
> 1. Fixed.
> 2. Learning is performed locally, and Assumption 1 specifies the variational distribution family $Q$, which is shared by all clients. On the server side, the aggregated posterior approximation also belongs to $Q$ because both WB and RKLB aggregation preserve Gaussianity and parameter-wise independence (see the “Personalization step” paragraph in Section 5.1). Importantly, this preservation arises from the aggregation rule itself, not from the Assumption 1. Adopting a common family $Q$ for the clients and the server also simplifies implementation and reduces computational cost.
> 3. *On the Gaussian Hessian.* For a multivariate Gaussian, the Hessian of the log-likelihood with respect to the variable is proportional to the inverse covariance matrix. Therefore, when the covariance is diagonal, the Hessian is also diagonal, allowing us to work directly with the corresponding scalar diagonal entries h and $\sigma$. *We added a footnote in the paper (page 5) to make this explicit.*
> 4. Fixed.
> 5. Proof updated as stated above.
> 6. Fixed.
> 7. *Choice of $\lambda$ vs. weights.*
> We intentionally use the parameter $\lambda$ because it provides a direct and intuitive connection to the personalization radius and leads to a clean inverse proportionality structure. Remark 3 motivates this choice and does not rely on an interpretation in terms of weights. That said, the equivalent weight-based formulation is mathematically valid, as the Reviewer noted.
> 8. *About personalization frequency.*
> Personalization is applied only once, after FL training has finished, and it does not affect the global model during training. Our work explicitly distinguishes three models - Local, Global, and Personalized - and compares personalized models against personalized baselines, as this is the intended focus of the method. For completeness, we also report the performance of the global model, and we show that our method achieves the best results on SVHN and CIFAR-10 across all three metrics.
> 9. *We do not fix the personalization parameter.*
> As clarified above, personalization is performed only once, at the end of training. Section 5.3 reports the performance of the personalized models for multiple values of $\lambda$, evaluated across clients. Section 5.4 then sets $\lambda$ = 1 as a reasonable trade-off that yields a balanced local–global behavior.

---

> ### Author Response · Authors · 2025-12-21
> **Authors Response - Visibility correction**
>
> Please accept our apologies. We mistakenly left our response dated 26 November with limited visibility in the system (visible only to the editors/authors). We have now corrected this. Our point-by-point responses and the corresponding manuscript changes are provided in that comment.

---

> > ### Comment · Reviewer_km55 · 2025-12-30
> > **Reply to authors**
> >
> > I thank the authors for the clarifications and for the updates to the manuscript. They have addressed the concerns that I previously had.

---

### Review · Reviewer_gqwS · 2025-12-01

**Summary Of Contributions:**

This paper proposes a novel approach to Personalized Federated Learning (PFL) based on an information-geometric projection framework for personalization in parametric Bayesian Federated Learning (BFL). The main idea appears to be original and (potentially) impactful; however, the empirical validation is performed solely on synthetic datasets, which makes the paper far less compelling then it could have been if it used real-world datasets that are available in the literature.

**Additional Comments:**

n/a

**Audience:**

Yes

**Audience Explanation:**

In general, Federated Learning has many practical applicabilities. However, the current evaluation on synthetic datasets does NOT raise to the standards of a journal publication, especially when real-world datasets are available (see the paper's reference [Tan et al, 2023])

**Claims And Evidence:**

No

**Claims Explanation:**

As the evaluation is done solely on synthetic datasets, it is unclear to what extent the results are truly meaningful and/or applicable in real-world scenarios. See suggestion below on evaluating on real-world datasets

**Requested Changes:**

I will begin by quoting two consecutive paragraphs from "Section V" in [Tan et al, 2023]:

>>> There are several FL benchmarking frameworks developed in recent years, including FLBench [75], Edge AIBench [76], OARF [77] and FedGraphNN [78]. LEAF [79] is one of the earliest and most popular benchmarking frameworks proposed for FL. At the time of writing, it provides six FL datasets covering a range of machine learning tasks including image classification, language modeling and sentiment analysis under both IID and non-IID settings. Examples datasets include the Extended MNIST [80] dataset split according to the writers of the character digits, the CelebA [81] dataset split according to the celebrity, and the Shakespeare [9] dataset split according to the characters in the play. A set of accuracy and communication metrics, along with implementation references for well-known approaches such as FedAvg, SGD and MOCHA are also provided. As LEAF extends existing public datasets from traditional machine learning settings, it does not fully reflect the data heterogeneity in FL scenarios. Although there are a few real-world federated datasets, such as a street image dataset for object detection [82] and a species dataset for image classification [83], they are often limited in size.

>>> Despite the release of benchmark datasets for FL, they are not widely adopted in PFL research. The vast majority of PFL studies choose to simulate the non-IID setting by performing their own partitioning on a public benchmark dataset used in machine learning (e.g., MNIST [84], EMNIST [80], CIFAR-100 [85]), or creating a synthetic dataset [17], [64], [68].

In this reviewer's opinion, this situation has become untenable: with the proliferation of PFL papers, it is simply unacceptable to continue evaluating the work on (semi-)synthetic datasets, especially when far more realistic ones are available [Tan et al, 2023]. The authors have a unique opportunity to perform a comprehensive evaluation on realistic/real-world data; if their results hold in these existing FL domains, they will set the bar against which all future approaches will be evaluated. If the current dataset are useful in making a particular point, they can be kept in the paper alongside the real-world ones.

OTHER COMMENTS:
- the authors should add 1-2 paragraphs explaining the relationships between the metrics used the paper and those from Table IV in [Tan et al, 2023]
- please add an intuitive paragraph discussing in-depth why the proposed approach does not perform on Fashion-MNIST as well as on the other two datasets
-

---

> ### Author Response · Authors · 2025-12-11
> **Authors Response**
>
> We thank the Reviewer for the constructive feedback. However, we would like to clarify the following points:
>
> **On the nature of the datasets (real vs. synthetic).** We would like to clarify a possible misunderstanding. The datasets used in our experiments (SVHN, Fashion-MNIST, and CIFAR-10) are standard real-world vision benchmarks, not synthetic datasets. The only “synthetic” aspect lies in the federated data partitioning: we simulate non-i.i.d. client distributions by applying a Dirichlet-based split to these datasets. This setup follows common PFL practice described in Tan et al. (2023) and is widely used as a standard benchmarking strategy in the FL literature, especially for cross-silo FL. Our work, motivated by reliability through Bayesian learning and interpretability through information geometry, targets applications such as hospitals and banks, which typically participate in cross-silo FL; *we have clarified this motivation and the corresponding choice of cross-silo simulations in the revised experimental section.*
>
> We also appreciate the Reviewer’s suggestion to broaden the empirical evaluation with additional federated datasets. The specific datasets mentioned by the reviewer are commonly used as cross-device FL benchmarks, with many clients each holding only a small local dataset (see Table 1 in [1]), whereas our study is explicitly focused on cross-silo scenarios with a small number of data-rich participants. A thorough cross-device study on multiple real-world benchmarks would therefore require substantial additional implementation to accommodate the different data modalities and heterogeneity patterns of the suggested benchmarks, as well as significant computational effort, particularly for the Bayesian baselines under comparison, and is not feasible within the present revision timeline. *In the revised manuscript, we explicitly state this focus and highlight a systematic evaluation on cross-device FL benchmarks as an important direction for future work.*
>
> [1] Caldas, Sebastian, et al. "Leaf: A benchmark for federated settings." arXiv preprint arXiv:1812.01097 (2018).
>
> **On the performance on FashionMNIST.** As shown in Table 2, on FashionMNIST our personalized model achieves the best performance on the global test data, improving accuracy by almost 4 percentage points over the second-best method, while remaining comparable to the baselines on local test data. Our global model also attains the second-best accuracy in both the local and global evaluations.
>
> We understand that these gains may appear less striking than the improvements observed on SVHN and CIFAR-10. However, we would like to emphasize two key points:
> - FashionMNIST is a relatively easy task on which all methods already achieve high accuracy. In this regime, strongly personalized approaches such as FedPop can overfit the local data, leading to poorer generalization on the global evaluation, whereas our method maintains a more favorable local-global trade-off.
> - The main claim of our work is not to maximize a single metric on one dataset, but to provide a method that yields a consistent balance between local and global performance across datasets and metrics. This is precisely what we observe in Figure 3, where our approach consistently lies near the Pareto frontier across the three datasets and three evaluation metrics.
>
> **On the metrics used and the comparison to Table IV in Tan et al.** We would like to clarify that, whereas our paper focuses on interpretable information-geometric perspective on Bayesian FL, Tan et al. (2023) is a survey that aims to cover many different aspects of FL. As a result, their taxonomy includes several families of metrics (e.g., system and communication performance) that are outside the scope of our work. Conversely, our study emphasizes metrics related to uncertainty quantification and calibration, such as ECE and NLL, which are not the main focus of the survey. We also note that Tan et al. (2023) primarily discusses the deterministic FL literature, while our contributions target the Bayesian FL setting. *We added a paragraph in the revised manuscript in Section 5.1 to make these differences explicit and to position our choice of metrics more clearly with respect to the categorization proposed in the survey.*

---

> > ### Comment · Reviewer_gqwS · 2025-12-17
> > **Thank you for the clarifications**
> >
> > Thank you very much for the detailed answers. I fully understand your arguments above, I agree that the 3 datasets are "synthetic" only wrt the federated learning aspect, and I apologize for my somewhat-careless comment that lead to the confusion. That being said, I find it difficult to accept that such a massive body of work (wrt federated learning) is based solely on datasets that are not real-world applications on TRUE federated learning domains. In my opinion, the field has far surpassed the level of maturity where at least one real-world dataset is used for benchmarking. Without such a dataset it is impossible to estimate the impact of a paper: is it making progress in the field? or is it just "overfitting" the quirks of the artificial split of one dataset or another?

---

### Decision · Action_Editor_6Dyx · 2026-01-01

**Recommendation:** Accept as is

**Audience:**

Yes

**Audience Explanation:**

Yes, the paper addresses a timely and relevant problem in Bayesian federated learning, and its theoretical and practical insights would be of clear interest to a substantial portion of the TMLR audience.

**Claims And Evidence:**

Yes

**Claims Explanation:**

The authors provide a solid theoretical foundation by showing, under mild assumptions, that the proposed information-geometric projection is equivalent to computing a barycenter on the statistical manifold, which justifies the claimed closed-form and cost-free personalization. This theoretical insight is complemented by a concrete algorithmic instantiation within a variational learning framework (IVON) and extensions to general aggregation schemes in Bayesian federated learning. Furthermore, the empirical evaluations under heterogeneous data distributions are well aligned with the claims.